

# A new 3D full-Stokes calving algorithm within Elmer/Ice (v9.0)

Iain Wheel[1], Douglas I. Benn[1,*], Anna J. Crawford[2,3,*], Joe Todd[1*], and Thomas Zwinger[4,*]

[1]School of Geography and Sustainable Development, University of St Andrews, St Andrews, UK
[2]Division of Biological and Environmental Sciences, University of Stirling, Stirling, UK
[3]School of GeoSciences, University of Edinburgh, Edinburgh, UK
[4]CSC - IT Center for Science Ltd., Espoo, Finland
[*]These authors contributed equally to this work and are listed alphabetically.

**Correspondence:** Iain Wheel (iw43@st-andrews.ac.uk)

**Abstract.** A new calving algorithm was developed in the glacier model Elmer/Ice that allows unrestricted calving and terminus advance in 3D. The algorithm used the meshing software Mmg to implement anisotropic remeshing and allow mesh adaptation at each timestep. The development of the algorithm along with the implementation of the crevasse depth law produced a new full-Stokes calving model capable of simulating calving and terminus advance across an array of complex geometries. Using a synthetic tidewater glacier geometry the model was tested to highlight the non-physical parameters that can alter calving. For a system with no clear attractor, model timestep and mesh resolution are shown to alter the simulated calving. In particular vertical mesh resolution had a large impact, increasing calving, as the frontal bending stresses are better resolved. However, when the system had a strong attractor, provided by basal pinning points, non-physical parameters have a limited affect on the terminus evolution. The new algorithm is capable of implementing unlimited terminus advance and retreat as well as unrestricted calving geometries, applying any melt field to the front, use in conjunction with any calving law or potentially advecting variables downstream.

## 1 Introduction

One of the largest sources of uncertainty in predictions of future sea level rise is the magnitude of losses from the Greenland and Antarctic Ice Sheets via ice discharge and iceberg calving (IPCC, 2023). In recent years, significant advances have been made in understanding calving processes and their relationship with ice dynamics (Benn and Åström, 2018), but this has yet to translate into the adoption of reliable, universal 'calving laws' in continuum ice sheet models. This largely reflects the contrast in the complexity of calving processes, which are influenced by stresses in three dimensions, and the simplified, vertically-integrated stress fields required in model simulations of long-term and large-scale ice sheet evolution. There is a need, therefore, to develop robust, physically based calving models in three dimensions, which can then be used to develop simpler calving parameterisations required for ice sheet models.

Currently there are two main types of 3D calving modelling methods with the capability to simulate the evolution of glacier calving fronts through time. The first is Discrete Element Modelling (DEM), which is commonly referred to as particle mod-





elling. An example model is the Helsinki Discrete Element Model (HiDEM) which predicts calving from first principles by treating the ice as elastically-bonded individual particles where the bonds fracture if a failure threshold is exceeded (Aström et al., 2014). HiDEM can accurately simulate a wide range of individual calving styles but is unsuitable for modelling longer-term glacier evolution because it does not include viscous deformation (Åström et al., 2013; van Dongen et al., 2020; Benn et al., in press). Conversely glaciers can be modelled as continua in 3D using a model such as Elmer/Ice. Elmer/Ice is a Finite Element Method (FEM) model that treats the ice sheet as a continuum and solves the flow and stress fields. Using the position-based crevasse-depth calving law, Elmer/Ice has been shown to accurately predict calving front evolution at Store Glacier (Sermeq Kujalleq) in response to changing fjord conditions (Todd et al., 2018; Benn et al., in press). Although the computational requirements of Elmer/Ice remain high, they are significantly lower than the requirements of HiDEM and multiple years of calving front evolution can be simulated with reasonable computational resources (Todd et al., 2018, 2019). Other methods use Linear Elastic Fracture Mechanics to predict calving (e.g., Krug et al., 2014).

However, several issues remained with using the Elmer/Ice calving model as implemented by Todd et al. (2019) to run multi-year simulations. Firstly, the corners of the ice-front in contact with the ocean were fixed in place. This is a reasonable assumption for stable glaciers such as Store Glacier (Sermeq Kujalleq) where only seasonal movement of the calving front needs to be captured (Todd et al., 2018). This assumption does not hold for fast-retreating glaciers such as Jakobshavn Isbræ (also Sermeq Kujalleq in Greenlandic) where the terminus position may change year on year by several kilometers (Joughin et al., 2020). Secondly, the Todd et al. (2019) model relied on a 2D vertically extruded mesh, which is problematic where the ice-front is non-vertical, such as where submarine melt produces undercutting (Todd et al., 2019). As a result, ice-front retreat leads to degeneracy of the mesh elements, which causes irrecoverable breaking of the calving model. Finally, Todd et al. (2018) assumed projectability of the calving front and calved icebergs, namely the post-calving ice front does not contain any complex re-entrants and can therefore be projected onto a transverse plane. This is clearly not always the case at real glaciers particularly those with complex front geometries such as Bowdoin Glacier (Kangerluarsuup Sermia). In order to rectify these limitations, a new calving algorithm has been developed. It utilises 3D remeshing which is implemented using the open-source remeshing algorithm Mmg (Dapogny et al., 2014). The level set method is used to define areas of calved ice (Osher and Fedkiw, 2001; Sethian, 1999) and the new calving front is physically implemented using Mmg. In this paper, the capabilities of the new algorithm are described and illustrated using a set of synthetic glacier geometries. Furthermore, often-neglected non-physical parameters with the potential to alter modelled calving are investigated, providing a sound basis for rational parameter choices in future studies using real world domains.

## 2 Modelling calving in a continuum

When modelling calving it is important to distinguish between the calving law, calving algorithm and calving model (Fig. 1). The *calving law* is the function by which calving is predicted, in this case the crevasse depth (CD) law (Nick et al., 2010; Benn et al., 2007). The *calving algorithm* takes the prediction from the calving law and implements this within the model,





ultimately leading to the removal of calved ice and the resulting alteration of the domain. Consequently, a calving algorithm is not tied to a particular calving law. Instead, it is the technical implementation, within a given ice-flow model, of a theoretical calving event provided by the independent calving law. Putting the calving law and calving algorithm together gives us the

*calving model*. This is an important distinction to make within glacier modelling, as both calving laws and calving algorithms are limited in their functionality at this time. Calving laws in 3D can be thought of as resulting from a combination of our current understanding of the physical processes behind calving and our ability to condense them into a mathematical function compatible with the ice-flow model. Calving algorithms can be viewed as the capability of our models to implement the calving law's prediction on a given domain by removing the predicted calved ice. This is surprisingly non-trivial especially for

more advanced calving laws that can produce convoluted calving geometries. This paper focuses on the development of a new calving algorithm that substantially increases our ability to realise calving in a 3D glacier model. This is primarily motivated by our previous limited ability to simulate substantial retreat and advance of fast-flowing glaciers in 3D. The objective is that once the previous technical hurdles have been overcome, our understanding and capability to test calving laws and physical processes can correspondingly be improved.


In this paper, we implement the CD calving law in the new algorithm. The CD law was chosen because it is the only currently available calving law that is explicitly based on physical processes, with proven capability in predicting calving on Greenland tidewater glaciers (Todd et al., 2019; Amaral et al., 2020; Benn et al., in press). The CD law is based on the idea that calving occurs when crevasses penetrate the full thickness of the glacier, or some prescribed part thereof (Benn et al.,

2007; Nick et al., 2010). For simplicity and ease of implementation, crevasse depth is predicted using the zero-stress approach introduced by Nye (1957) and modified by Todd et al. (2018), which assumes that fracturing is possible wherever the largest principal stress $\sigma_1$ (the largest eigenvector of the Cauchy stress) is extensional (positive). This approach assumes negligible stress concentrations at crack tips, which is reasonable for fields of closely-spaced crevasses. Calving is thus a function of the large-scale stress field of the glacier, particularly regions of high extensional stress. It is important to emphasise that the CD law

does not predict individual crevasses or impose discontinuities on the model glacier. Rather, the zero-stress function defines bounding surfaces of crevasse fields (i.e. the base in the case of surface crevasses fields, the top in the case of basal crevasses fields), which are then used to predict the position of the glacier front at a particular timestep.

## 3   Implementation of calving in a continuum model

Implementing any calving law in a continuum model requires the new position of the ice front to be prescribed at each timestep.

In vertically-integrated models this is relatively straightforward because the ice front is by definition vertical before and after calving and the model physics can be solved on a 2D mesh. The problem becomes substantially more difficult in three dimensions when the front can have complex vertical profiles due to melt-undercutting or deformation.

Representing the change in the glacier domain via calving can either be done by an explicit change in a domain (i.e., modifica-



# Calving model

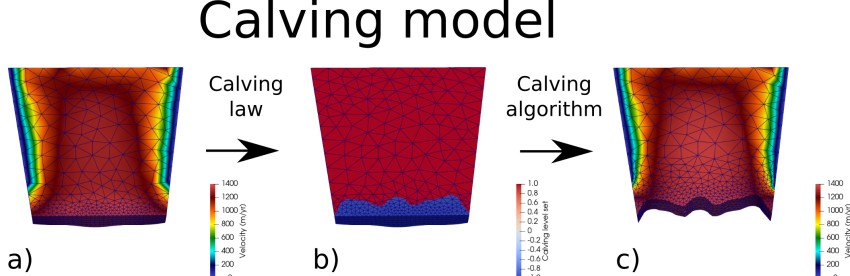

**Figure 1.** Illustrating the difference between the calving model, algorithm and law. a) The initial flow solution. The calving law then predicts calving based off the solved variables on the initial geometry. b) The predicted calving based on the calving law. This prediction is in the form of a level set variable which is then fed to the calving algorithm. Nodes with a negative value (in blue) show areas to be calved and those with positive values (in red) show the remaining glacier domain. c) The new geometry with all the variables interpolated across showing the interpolated velocity field. The calving model represents the combination of both the calving law and algorithm allowing you to go from a) to c).

tion of a mesh) or by using a 'calving variable' that represents the new terminus position. Computationally it is much cheaper to use a variable since only this variable needs to be manipulated prior to calculation of the new flow solution. This approach is becoming increasingly popular, especially among lower-level models where computational efficiency is the priority (e.g., Bondzio et al., 2016). Usually this is done using a level set method (Osher and Fedkiw, 2001; Sethian, 1999; Bondzio et al., 2016) where a surface is defined from a signed distance and moves based on an advection equation. The main drawback of this

method is the need to solve intra-element dynamics if a level set surface is to be followed exactly since the hyperplane will cut through any form of element. No current model has the capability to solve this problem, which would require boundary conditions to be applied across the hyperplane (e.g., Bondzio et al., 2016). Instead, intersected elements are marked, those beyond are marked as ice free and those in the glacier domain marked as ice. Boundary conditions are instead applied across the ice-free boundary of the intersected elements, so the actual modelled calving front can differ greatly from the level set

surface (Bondzio et al., 2016). Ice-free elements remain dormant.

Alternatively, explicit domain change is more computationally expensive and significantly more complicated to implement. Simplistically, this can be achieved by deforming the mesh (i.e., moving nodes but keeping elements intact), but this can only be done for non-complex geometries and becomes less feasible with more dimensions. Complex problems require complete

remeshing where element edges are realigned along the new calving front. Todd and Christoffersen (2014) produced a flowline 2D model that allowed vertical changes across the terminus, but concluded that all three dimensions needed to be considered for accurate calving representation. A similar approach was taken by Berg and Bassis (2022) where the vertical dimension was seen as imperative to the study of calving dynamics and ice history. The addition of a third dimension makes modelling significantly more complicated. The most advanced example is the 3D extruded remeshing implemented by Todd et al. (2018).





Several advantages emerge from this approach. For example, complete remeshing allows finer elements to be maintained near the calving front if the terminus moves greatly over the course of the simulation. Secondly, the position of the calving front resembles that defined by the calving law much more closely, allowing implementation of more complicated position-based calving laws.

## 4    The new calving algorithm: methods and capabilities

In this paper, we illustrate the features and capability of the new algorithm using a simple synthetic geometry broadly representative of a Greenlandic glacier but with reduced computational costs. The synthetic glacier is dominated by strong shear margins with super buoyant sections of the terminus. All of the experiments in this paper were completed on a local desktop on eight processors. The full geometry and input files can be found in the official Elmer/Ice repository (https://github.com/ElmerCSC/elmerfem) or in Wheel (2023a) in the test case elmerice/Tests/Calving3D_lset. Full details of the as-
sociated geometry and boundary conditions are detailed in Appendix A. Rather than give the unrestricted details and coding design associated with the algorithm, the key methodological choices and model capabilities are presented here highlighting their benefits and drawbacks. A comprehensive breakdown of the algorithm and minor coding choices is provided in the supplement or in Wheel (2023b). Detailed user documentation is provided in the Elmer/Ice repository or in Wheel (2023a).

### 4.1    Explicit domain modification

The domain of a glacier actively evolves through two mechanisms: the movement of the glacier terminus via ice flow along with frontal melting and retreat of the terminus via iceberg calving. Additionally, the elevation of the surface and base of the glacier can evolve through ice flow, melting or accumulation. The implementation of the movement of the terminus, base and surface boundaries is achieved through deforming the mesh, while calving is implemented through remeshing.

### 4.2    Mesh deformation: terminus advance and free surfaces

The Lagrangian advection of the terminus for a given timestep is computed using the velocity solution. For constricted domains such as glaciers flowing down a fjord, the advance of the lateral margins must account for local geometry. Across the majority of the terminus, the displacement vector ($\boldsymbol{d}$) is the velocity multiplied by the timestep. However, where the calving front and the lateral boundaries meet, the displacement vector cannot be based purely on the velocity since the terminus advance must be constrained by the fjord walls. This could be solved as a contact problem analogously to the grounding line (Durand et al.,
2009), but this would increase computational requirements as the non-linearity of the problem increases. Instead, fjord walls are user defined before the simulation and the velocity is projected along that margin. As such the displacement in the x and y plane at the lateral margins ($\boldsymbol{d_l}$) is defined by

$$\boldsymbol{d_l} = |\mathbf{u}| \times \boldsymbol{f}, \tag{1}$$





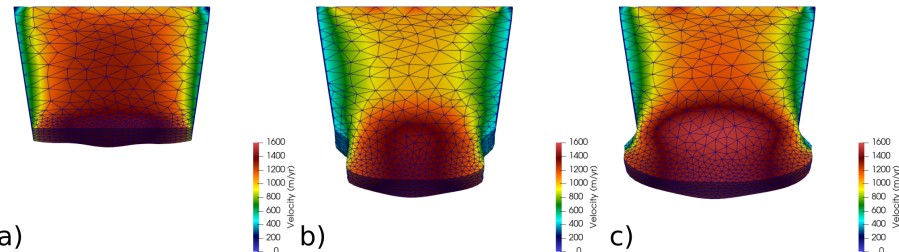

**Figure 2.** Two examples showing the capabilities of the front advance routines over one year simulations. Calving is suppressed for both simulations so the resultant geometry is purely from glacier advance. a) The initial geometry prior to simulation. Final geometry and velocity field for b) a fjord with a narrowing and c) a widening fjord. The full simulations can be seen in Supplementary Videos 1 and 2.

where $\boldsymbol{u}$ is the velocity and $\boldsymbol{f}$ is the direction of the fjord wall. Although this is very computationally efficient, it may lead to
artificial mass change as the imposed kinematics of the lateral boundary corners are not guaranteed to obey the incompressibility condition. This artificial mass change is negligible, however, as the velocity remains parallel to the lateral margin because of the standard lateral boundary condition of zero normal velocity. The only location where velocities are not parallel to the lateral margin are the lateral boundary corners. This is because the normal vectors are not uniquely defined at these locations. This provides a much better solution than having the lateral corners fixed in place and time (Todd et al., 2018). The mesh
is deformed horizontally using the predicted front advance as a boundary condition. Similarly, the kinematic free surface is calculated for the surface and base before the mesh is deformed vertically using these solutions. This accounts for the glacier surface mass balance, while grounding-line evolution is solved as a contact problem (Durand et al., 2009). In order to simulate terminus advance through complex fjord geometries, the model has the capability to transfer boundary elements from the terminus to the lateral margins. The model can replicate realistic advance at widening and narrowing fjords whilst maintaining
the appropriate boundary assignment (Fig. 2).

Correspondingly to the terminus advance, the retreat due to submarine melting can be applied to the calving face. It is applied as a scalar variable where the direction of melt is always assumed to be normal to the terminus. Following the Lagrangian implementation of the glacier advance, melt is prescribed such that

$$\boldsymbol{d} = \boldsymbol{u} - \hat{m}, \tag{2}$$

$$\boldsymbol{d_l} = |\boldsymbol{u}| \times \boldsymbol{f} - \hat{m}, \tag{3}$$

where $\hat{m}$ is the melt normal to the front. Melting with any vertical or horizontal profile can be implemented and the calculation of which is independent to the calving algorithm. Given the simplicity of this method there are very few issues that can arise
when applying melt. Degenerate elements will only be produced if the melt per timestep is larger than the element length in the normal direction.





### 4.3 Calving through remeshing

The calving algorithm is defined as the implementation of calving, or the removal of ice from the glacier front. It takes a level set or signed distance variable where the zero contour is the new calving front to produce a new mesh with all model
variables interpolated across onto it. Importantly the new calving algorithm is not limited by iceberg or frontal geometries and consequently is not tied to a particular calving law. Given its physical nature and use as a position-based law, the CD law is a particularly complex calving law to apply in a 3D model. It therefore provides a high benchmark and simpler rate based laws could easily be applied. The CD law is implemented following Todd et al. (2018) but improvements allowing non-projectible calving have been made through use of a level set function (Osher and Fedkiw, 2001; Sethian, 1999). For further details see
the user documentation detailed in the Code and Data Availability.

The other major issue with the previous Elmer/Ice calving algorithm was element degeneracy for certain geometries. These issues stemmed from the use of a vertically extruded mesh (Todd et al., 2019). Once the calving variable had been defined, the mesh was compressed or stretched to match vertical terminus geometry changes. This was particularly problematic when
submarine melting was applied leading to a non-vertical terminus. In such geometries an average front position was determined from which a 2D footprint was created and meshed. Several filters had to be applied to move and delete nodes to reduce the prevalence of degenerate elements. The footprint was extruded vertically and then deformed to match the front geometry of the old mesh. To overcome these issues, the remeshing software Mmg is used in the new calving algorithm to produce a fully 3D domain without the need for vertical extrusion (Dapogny et al., 2014). To use the calving algorithm Mmg (version 5.5.4
or later) must be compiled along with Elmer. The major limitation of using Mmg is its requirement to run in serial, vastly reducing the scalability of the calving algorithm. However, large-scale testing has shown this is not an insurmountable problem at present, as solving the Stokes equations is still the major computational requirement for any simulation.

  Currently, remeshing is completed in two separate steps. The first step realigns element edges along the zero level set contour.
This will be referred to as implementing the level set variable. The second stage is complete anisotropic remeshing to improve the mesh quality. The full remeshing algorithm is visualized in Fig. 3 and outlined in Fig. 4. To reduce the computational requirements, only the area within a user defined distance from the terminus is remeshed. Unfortunately, Mmg must run in serial so that in a parallel run the Elmer mesh partitions must first be gathered onto one process (Fig. 3b-c). Essentially, for both remeshing stages, the nodes and elements on the upstream partition boundary of the gathered mesh are fixed and cannot be
altered. This means that when converted back into the Elmer mesh format, the new mesh still has the same partition boundaries with the upstream parts of the mesh which have not been altered. As the current implementation of remeshing using Mmg must run in serial, the gathered mesh must be redistributed using the library Zoltan (Devine et al., 2009). A rebalancing algorithm aims to rebalance the mesh evenly in terms of computational requirements among all active processes whilst trying to reduce parallel communication (Fig. 3f). Additionally, the ability to use ParMetis for rebalancing is also possible, and provides much
better balancing for larger jobs. After the rebalancing, variables from the old mesh are interpolated across the new mesh in





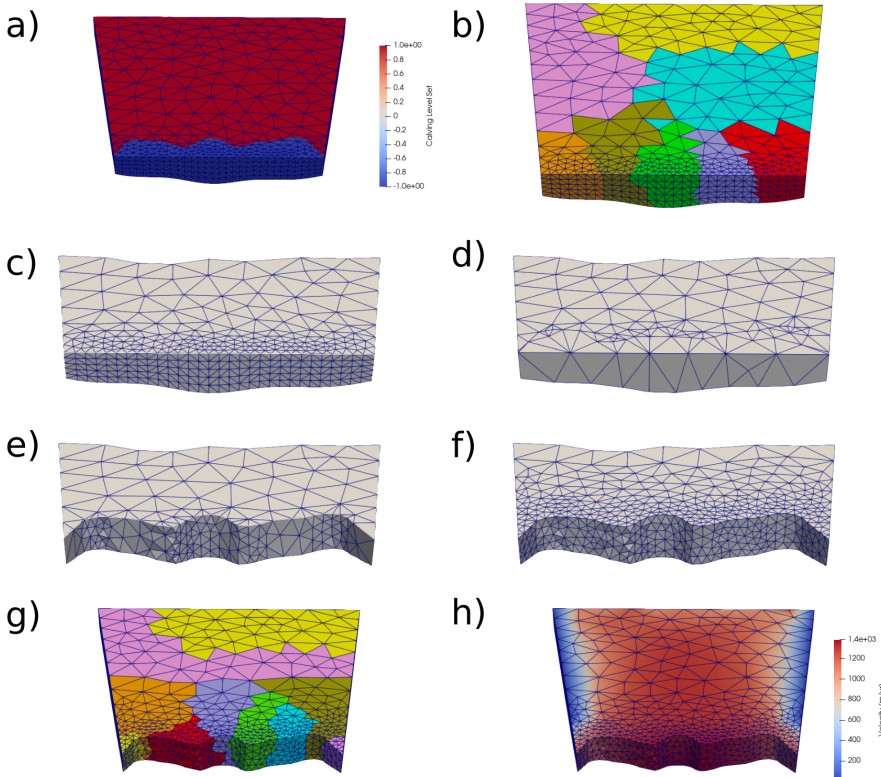

**Figure 3.** A visualisation of the steps involved during remeshing in the calving algorithm for a simulation run on eight cores. a) The 3D glacier mesh showing the calving level set variable defining the predicted calving. The reduced range of the calving level set variable is just to clearly show the predicted calving. b) The same distributed mesh showing the eight partitions. c) The gathered mesh on one core. The upstream size is defined by the user defined remeshing distance from the new calving front. d) The first remeshing step where element edges are aligned along the new calving front as predicted by the calving level set variable shown in a). e) After the first remeshing step the nodes with negative calving level set values are cut (e.g. glacier ice that would calve as icebergs is removed). f) The second remeshing step where the mesh quality is improved. g) After remeshing the mesh is rebalanced so each partition has a roughly equivalent partition. h) The variables are interpolated to the new mesh in parallel and the old mesh is deallocated from memory.

parallel (Figs. 3, 4). Surface and bottom variables are projected from the old mesh as both surfaces must maintain projectability. This is assumed as free-surface solvers are used on both. The free surface cannot be solved on a non-projectable boundary. However, with the possibility of complete remeshing, there are occasionally sections of the surface and bottom boundaries that are not covered by the old mesh. Nodes here are individually extrapolated.





## 4.4 Robustness of algorithm

Given the complexity of potential geometries arising from calving at a tidewater glacier, it can be expected that instances of remeshing failure will occur during simulations. It is not possible to prevent this for every scenario that may arise, so additional focus must be placed on the robustness of the calving algorithm to cope with remeshing failure. A major advantage of performing the level set variable implementation and anisotropic remeshing in two steps is the ability to isolate a source of potential failure. If there is an issue, remeshing can be rerun without affecting the level set variable implementation. If a failure occurs during the level set variable implementation or anisotropic remeshing, the process is retried with finer mesh input parameters. This often results in success (Fig. 4). Failure can occur for several reasons.

The first reason that failure occurs is that Mmg is unable to return a saved mesh. Secondly, remeshing failure can occur if the element quality does not meet the user defined minimum quality. However, there can be times when level set implementation fails across the range of input parameters provided. Calving cannot occur in this case. Remeshing still occurs to try to improve the mesh quality. Similarly, if remeshing fails on all input parameters, calving does not occur even if the level set variable implementation has been successful. Both situations can lead to the glacier falsely advancing. This is not a major issue as calving will likely occur on the subsequent time steps.

A final potential issue with the remeshing can result if a mesh passes through the various quality checks but has some physical imperfection or poor element quality that leads to problems in the Stokes solver. Element quality checks attempt to prevent such instances but are not fool proof. Such imperfections often cause unrealistic velocity solutions that in turn can cause unrealistic calving events. An additional step in the calving algorithm has been added to check the new flow solution, which ensures that a mesh imperfection does not slip though. Finally, model check pointing, by regularly saving the model state, allows for easy recovery.

If the flow solution is determined to be inadequate (non-converged), the mesh deforming and calving solvers are suppressed except the anisotropic remeshing routines. The mesh input variables are refined and the flow solution residuals are removed, so that the velocity is calculated from scratch. These refinements result in a finer mesh and an extension of the remeshing distance. The latter is extended in case the fixed elements are causing the flow-convergence issue. Additionally, the model time is set back to the time at the start prior to the current, problematic, time step plus one second (Fig. 5). An extra time step is added to the required time steps for the simulation. Following the subsequent remeshing step the mesh quality will usually have improved enough to provide a flow solution. At this point, the solver will unpause the mesh deforming and calving solvers and return the mesh input variables to their original values.





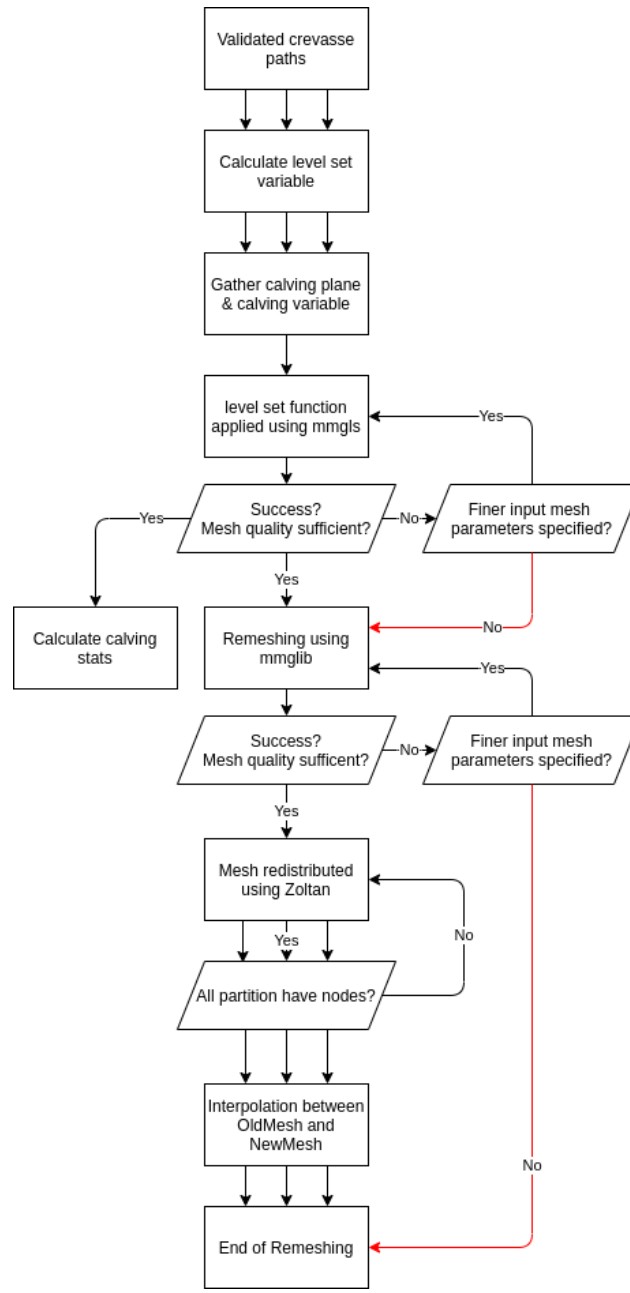

**Figure 4.** Diagram outlining the steps involved in the calving algorithm. A visual representation of an example simulation can be seen in Fig. 3. Red arrows indicate paths where calving is suppressed. Single arrows indicate stages which occur in serial and three arrows indicate stages that can occur in parallel.





## 5  Typical simulation

A typical simulation is now outlined, putting together the new advance, calving and remeshing mechanisms (Fig. 5). After model initialization and set up, the solvers follow the algorithm. First, the velocity field is solved. It is then checked to assess it for any abnormalities. If abnormalities exist, the recovery mechanisms outlined above are followed.


Assuming the flow solution converges, it is used to solve the stress fields from which calving is predicted by the CD law. After the calving prediction, the top and bottom free surfaces are solved using the built-in Elmer free-surface solver. As outlined previously the terminus advance is calculated from the flow solution. Using these variables the glacier mesh is deformed both vertically and horizontally. Ideally, the flow solutions would be recalculated after the front adjustment since the calculated

velocities are based on a different geometry. This means that, currently, the calving prediction is based on the stress field that was calculated for a slightly different geometry. This purely explicit in time approach is used currently to save computational requirements. Unless timesteps are extremely large for the size of domain this should not impact the results.

After the mesh deformation stage, the calving algorithm is called. If remeshing is unsuccessful at any stage, calving is sup-

pressed and the model moves onto the next time step. Following recovery through successful remeshing, any solvers that were previously paused are turned back on and the timestep is checked to make sure it is the original input. If remeshing is initially successful but only insignificant calving occurs the timestep is not altered. However, if an iceberg calves above the user defined threshold, mesh deforming solvers are paused, the timestep is reduced and an additional timestep is added to the model to allow the simulation to run for required time (Fig. 5).

## 6  Model parameter experiments


Modelling calving is dependent on not just the physical set-up but also any defined model inputs. These do not include tuneable physical parameters that are often adjusted to improve results (e.g., crevasse penetration requirement, Glen Exponent, Von Mises stress threshold). The defined model inputs are instead parameters unrelated to the physical environment and exist purely within the model domain. Despite this, they can alter the way the model captures physical processes such as calving.

Conceptually, the CD law predicts the attractor within the fluctuations of terminus advance and retreat (Benn et al., in press). In order to best show how non-physical factors within the numerical system can alter the predicted calving, two synthetic domains were created. The first has a sloping bed with no bed irregularities producing a system lacking a strong attractor. The second has two basal rises (i.e. potential pinning points) near the initial terminus position which act as an attractor within the system. The model setup is described in Appendix A. The control value for each parameter is described at the start of the relevant

section. Mean retreat rates were calculated as the mean displacement of the terminus per timestep along the y-axis.



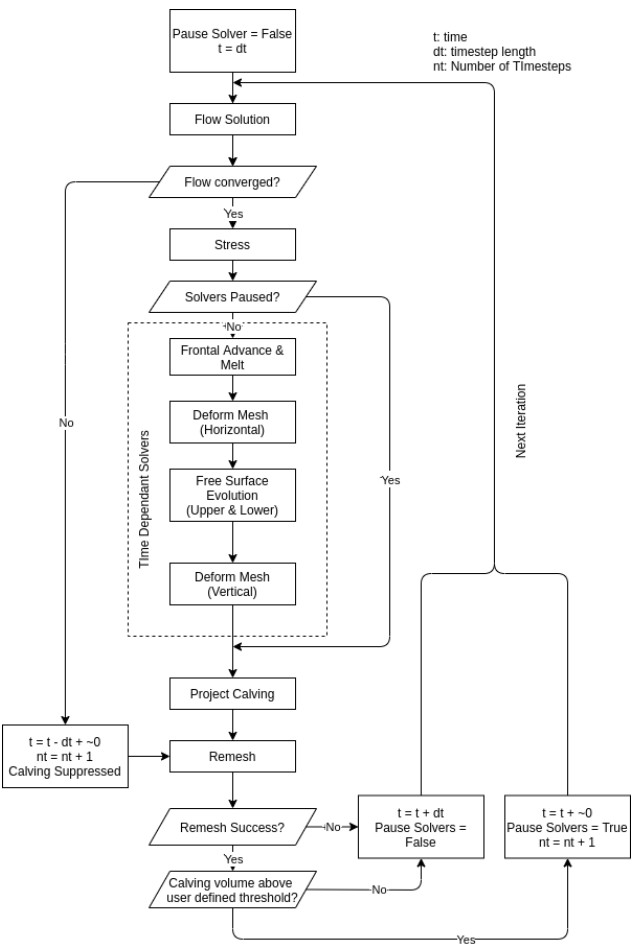

**Figure 5.** Diagram showing the stages involved in a typical simulation using the calving algorithm and front advance routines in tandem.

## 6.1 Timestep magnitude

Like any computer-based model, FEM requires a model timestep that represents the time gap between the calculation of the model physics. When solving glacier flow, a larger timestep reduces the number of times the Stokes equations must be solved, reducing the computational cost. However, too large a timestep can cause too much divergence in the flow solver either through
the occurrence of instabilities or reduced accuracy. This is especially true when working with an evolving domain such as an advancing or retreating glacier. Usually, the timestep must be small enough to meet the Courant-Friedrichs-Lewy (CFL) condition (Courant et al., 1928) and ensure that the maximal velocity displacement over the timestep is smaller than the minimum grid size. Therefore, the timestep size is often the minimum value which allows convergence of solvers based on the glacier velocity and mesh size where the transport of properties imposes a CFL citerion. The use of a position-based calving law,
which calculates the ice-front location from an instantaneous state of the evolving glacier geometry and stress field, means





its solution could be affected by the assigned timestep. Rate-based laws reliant on the changes in the stress field such as the Von Mises calving law (Morlighem et al., 2016) will also be timestep dependent. In contrast, a non-physical rate-based law is timestep-independent so would not be affected in the same way by changes in the assigned timestep.

Four additional simulations with timesteps of 2d, 0.55d, 0.33d and 0.25d were run to complement the control simulation with the timestep of 1d (Fig. 6a, b). All the simulations ran for a total of 100d. The initial calving event on the first timestep remains similar irrespective of assigned timestep. A reduction in the timestep leads to continual retreat when no pinning points are present. There is no clear convergence with each reduction in the timestep leading to more retreat. In contrast, when pinning points are present, the system converges at 0.33d. Further reduction of the timestep to 0.25d does not yield further retreat.

Notably, the control experiment of 1d and all the experiments with a reduction in the timestep show very similar terminus positions and shapes (Fig. 6b).

The increased calving modelled when smaller timesteps were applied can be explained by the increased number of times that calving was predicted. As the timestep reduces, the frequency of computing the stress field on a unique geometry in-

creases. This increased frequency of calving prediction increases the modelled calving because it heightens the probability of successful calving. For a glacier system with no strong attractor, calving is highly dependent on the timestep. When an attractor is added, in the form of pinning points, the predicted calving is much less sensitive to the model timestep. Diminished returns are seen here, and the increased retreat converges at 0.33d.

## 6.2   Adaptive timestep

The calving algorithm can add additional, shorter timesteps if large calving events occur to determine whether the calving-induced change in geometry will lead to further, immediate calving (Fig. 5). The control simulation could add up to three additional small timesteps of $1 \times 10^{-10}$ yr if the calving threshold, set as $1 \times 10^{7}$ cubic meters, was reached at the prior timestep. Two further simulations were done for each domain, one with the adaptive time stepping deactivated and one in which up to five additional timesteps could be added (Fig. 6k, l). The calving threshold at which the adaptive time stepping was invoked and the small timestep size were not changed from the control.


Without the adaptive timestep the terminus retreated at a slower rate but ultimately reached the same stable point as the control regardless of the domain (Fig. 6k, l). The positions for the more sensitive domain without pinning points are shown (Fig. 6k, l) but when pinning points are present the ultimate terminus also matches the control. Conversely, with more adaptive

timesteps the terminus retreated at a quicker rate from the unstable starting geometry. Over time, the terminus positions slowly converged towards the same position where at 25d they are almost identical. From here on the terminus positions do not diverge.

Adaptive time stepping intuitively makes sense based on our knowledge of secondary calving and so is important if a timeseries of terminus positions is wanted. If instead only the ultimate terminus position for a longer-term simulation is required, it is



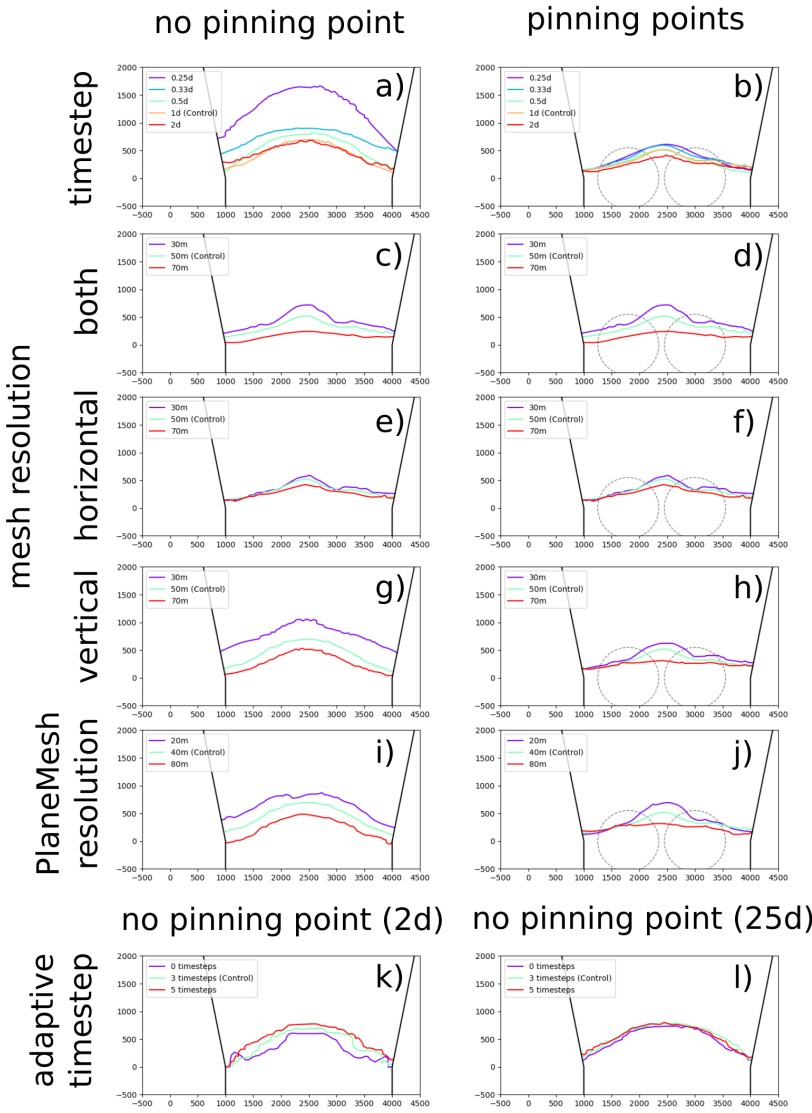

**Figure 6.** Terminus positions at the end of the non-physical experiments. a, c, e, g, i, k, l) were run on the domain with no pinning points and b, d, f, h, j) with basal pinning points. The non-physical parametered altered for each simulation shown were: a, b) timestep magnitude, c, d) horizontal and vertical mesh resolution, e, f) horizontal resolution, g, h) vertical resolution, i, j) PlaneMesh resolution and k, j) adaptive timestepping. The mean retreat rates are presents in Table 1.




unlikely that adaptive time stepping will change the result. Therefore, when using the adaptive time stepping functionality, it is important to consider the topic of investigation and timescales of interest for the simulation.

### 6.3    Glacier mesh density

As previously stated, choice of mesh density is by the CFL criterion corresponding to the assigned timestep in glacier simulations. Increased horizontal and vertical resolution can refine the flow and stress solution, especially at areas of high shear. It
is for these reasons that ice-sheet scale simulations often have fine meshes at ice streams (e.g. Gillet-Challet et al., 2012). In the simulations shown in this paper, the smallest elements were closest to the glacier terminus where bending and extensional stresses are prominent and essential for determining calving locations. Increased mesh density comes at a computational cost, roughly increasing exponentially as the resolution increases. This is because doubling the resolution quadruples the number of elements in the mesh. Usually mesh density is dictated by computational resources, but the small synthetic geometry used in
this testing allows us to further investigate changes in the mesh resolution.

Experiments with varying mesh densities were conducted for both domains (Fig. 6e-h). The control had vertical mesh resolution of 50m both vertically and horizontally at the terminus, and additional simulations were run with finer mesh resolution of 30m and a coarser resolution of 70m. Vertical and horizontal mesh resolution were investigated independently and collec-
tively producing six experiments for each domain (Fig. 6e-h).

Increased glacier mesh density leads to full terminus retreat when no pinning points are present (Fig. 6e). When pinning points are present the retreat is limited to where the glacier is pinned. A reduction in the glacier mesh density limits retreat on both domains. Independently changing the horizontal resolution has a limited impact on calving in either domain. Vertical
mesh resolution has a far larger impact accounting for most the retreat seen when mesh resolution is increased across all planes (Fig. 6g, h).

The vertical resolution had a much larger impact on calving and it is important to understand why. The vertical resolution alters the penetration of basal crevasses as – depending on the resolution value – more or fewer elements are present near the
waterline. Greater resolution allows the ice-cliff imbalances and bending stresses to be resolved more fully, as the ice close to the waterline is depicted in more detail (i.e., more nodes are present around the waterline). In the simulations with finer resolution, multiple elements are present above the waterline but in the coarser mesh only one node is above this point. If the ice-cliff imbalance and bending stresses above the waterline are resolved in more detail, more extensional stress is captured above the waterline. Consequently, principal stress values are higher allowing surface crevasses to penetrate more of the ice
column. Similar retreat to experiments shown here could be replicated by just increasing the vertical resolution 100m either side of the waterline. The importance of resolving the ice-cliff imbalances and bending stresses highlights two key details that need to be considered when applying the CD calving law. The first is that resolving the flow using the full-Stokes flow is essential so the bending stresses can be accounted for. The second is that vertical mesh refinement could potentially be





valuable for reducing computational costs while successfully simulating calving dynamics. This would follow in a similar vein
to the understanding of the horizontal anisotropic remeshing being important in ice-sheet scale modelling to resolve the flow
dynamics at ice streams (e.g., Gillet-Chaulet et al., 2012).

### 6.4  Plane mesh density

When calving is predicted the crevasses are mapped onto a 2D mesh, known as the PlaneMesh, the resolution of which is
independent of the 3D glacier mesh. Since there are no partial differential equations being solved on it, the resolution of the
PlaneMesh does not greatly change the computational cost of the algorithm. Its use, however, is not fully parallel so does not
scale as well as the 3D glacier mesh.

For each domain, two simulations with PlaneMesh resolutions of 20m and 80m were run for comparison against the con-
trol that had a resolution of 40m (Fig. 6i, j). Increased PlaneMesh density led to more calving particularly when no pinning
points were present. Here, increased PlaneMesh resolution showed a linear increase in calving with no clear convergence pat-
tern. When pinning points were present the retreat was limited when the resolution was increased. For the coarser PlaneMesh
resolution of 70m the terminus undergoes limited retreat and diverges from the compressive arch shape seen in the other sim-
ulations.

Increasing the PlaneMesh resolution increases the number of points at which the vertical ice column is assessed for the compu-
tation of crevasse penetration (Fig. 6 i, j). This can increase calving by two mechanisms. Firstly, by reducing the space between
these points, the location of crevasses inducing calving can be more accurately determined. This will often shift the crevasses
up glacier slightly when the modelled crevasse location is between PlaneMesh elements. The maximum upstream refinement
in crevasse location is determined by the reduction in element size. Secondly, increasing the resolution of the PlaneMesh, by
increasing the number of nodes, increases the likelihood of successfully isolating a crevasse contour.

The low computational cost of the PlaneMesh routine means that a fine mesh resolution should always be used. It is diffi-
cult to determine how fine a resolution to apply, especially when there is almost certainly a relationship with the 3D glacier
mesh resolution as well. The lower change seen when the horizontal glacier mesh resolution is altered suggests a dependence
on PlaneMesh resolution that warrants further investigation. The resolution of crevasse mapping does affect calving and so
the sensitivity to a given set-up should be considered in future glaciological applications of the algorithm. The PlaneMesh
resolution should remain consistent between experiments as well.

### 6.5  Summary of the influence of non-physical parameters on calving

In summary, this set of experiments shows that changes in timestep length, mesh density, and plane mesh can affect predicted
calving in the new algorithm. Importantly, however, changes in these parameters have a much smaller impact on the predicted
ice-front position when a pinning point is present. That is, the model converges on similar solutions for all chosen values of





**Table 1.** Summary of the mean retreat rates for 100d experiments testing the non-physical parameters that can alter calving.

| | | Mean retreat rate (m/d) | |
|---|---|---|---|
| Non-physical parameter | Experiment | No pinning points | Pinning points |
| Control | | 3.31 | 4.36 |
| Timestep | 0.25d | 3.67 | 12.16 |
| | 0.33d | 3.57 | 7.27 |
| | 0.5d | 2.99 | 5.50 |
| | 2d | 2.51 | 4.52 |
| Mesh resolution | 30m | 4.39 | 9.12 |
| (both) | 70m | 1.57 | 2.03 |
| (vertical) | 30m | 3.86 | 7.72 |
| | 70m | 2.48 | 2.89 |
| (horizontal) | 30m | 4.35 | 5.66 |
| | 70m | 2.63 | 3.94 |
| PlaneMesh | 20m | 3.89 | 6.29 |
| | 80m | 2.40 | 2.33 |
| Adaptive time | 0 timesteps | 3.11 | 4.18 |
| | 5 timesteps | 2.88 | 4.45 |

parameters when the system contains a strong attractor. In contrast, when such an attractor is absent, the model runs simulate different rates of retreat depending on parameter choices. This suggests that the choice of temporal and spatial model resolution is of greatest importance where transient glacier behaviour is of interest.

## 7 Summary of a model capabilities and potential

Putting together the new calving algorithm along with the upgrades in the calving projection gives us a new model with unparalleled capabilities of simulating calving over a 3D continuum. New features of the model include:

1. Unlimited advance or retreat now possible to simulate in 3D.

2. Unrestricted 3D calving geometries can be utilised by the model.

3. Any calving law can be implemented

4. Features or variables can be advected as part of the mesh.

5. Any 3D melt field can be applied to the glacier front.





## 7.1 Unlimited retreat and advance

The ability to model unrestricted retreat and project advance in 3D is a major step forward for simulating the dynamics of
calving glaciers. Previous 3D calving models have been limited by technical hurdles that prevented retreat or advance beyond a
certain location (Todd et al., 2018). In the most extreme warming experiments conducted for Store Glacier the simulated retreat
caused model breakdown (Todd et al., 2019). The model breakdown prevented a full comparison of calving dynamics through
the perturbation of the input variables. A second important opportunity made available with unlimited retreat and advance is
the ability to model any glacier or ice shelf in the world. The Todd et al. (2019) model was limited to stable glaciers that do not
undergo large seasonal or interannual variability.

## 7.2 Unrestricted calving

The ability to calve unrestricted geometries of icebergs both in the horizontal and vertical planes should be treated as a distinct
feature of this calving algorithm. This means any potential configuration of calving or front geometry is possible. Again, this
ability opens up the possibility to model any complex scenario or situation seen in the real world. For example, glaciers with
complex front geometries such as Bowdoin Glacier (Van Dongen et al., 2019) or large fan-shaped ice tongues which are non-
projectable can now be modelled. The new calving algorithm also offers the possibility to create unrestricted synthetic calving
geometries to explore how the glacier dynamics respond to forced calving events. In the same vein, other calving laws could
be implemented. Their implementation would not be restricted by possible calving predictions, no matter how complex the
resulting geometry.

## 7.3 Use of other calving laws

The calving algorithm is not restricted to the CD law implementation outlined above. Any calving law could be implemented
through the production of a level set variable or signed distance variable that is given in the calving algorithm. Despite the
relative ease by which a new calving law could be implemented, only the CD calving law has been used up to now. This is
because it is currently the only possible option based on physical processes (Benn et al., 2017). Other popular calving laws
are based on calving rates as opposed to calving position and they could be used in conjunction with this calving algorithm.
The calving algorithm provides an easily accessible framework for which various calving laws could be compared in 3D.
More likely, alterations or improvements will need to be added to the CD law as coupled modelling of tidewater glaciers
advances (Cook et al., 2023). Possible ways in which the CD law could be developed include the incorporation of ice history
via advection of damage, and implementing more sophisticated methods of calculating crevasse depths. However, this new
algorithm provides the best framework to approach these problems, as 3D modelling is no longer restricted by technical
hurdles. As such we can now focus on improving the calving laws along with assessing which missing processes are important
in calving prediction.





## 7.4 Feature advection

The advances in remeshing techniques allow complete anisotropic remeshing of a particular glacier part or complete glacier.
This allows mesh quality to be maintained even if nodes are moved in a Lagrangian manner. Some issues remain related to element degeneracy at the lateral margins, but this does not apply to ice shelves such as those extending from Thwaites Glacier which are laterally unconstricted (Scambos et al., 2017). There is very exciting potential for use of the remeshing techniques to advect variables such as damage downstream in order to better predict calving, especially on ungrounded ice sheets (Cook et al., 2023).

## 7.5 Use for melt simulations

Beyond the calving component of the new algorithm, the ability to model the effects of submarine melt on glacier dynamics in 3D is very novel. The limited availability of full-Stokes glacier models mean the effects of 3D melt fields on glacier dynamics are rarely researched. The ability of the algorithm to have a non-projectable front means any melt field could be applied for any length of time without model breakdown. More commonly, the importance of submarine melt is often associated with its known
ability to increase calving (O'Leary and Christoffersen, 2013). Melt undercutting from buoyant plumes is often connected to full terminus retreat through calving, especially at the lateral margins (Cowton et al., 2019). Investigating the importance of submarine plume melt undercutting on calving is only possible in a 3D calving model because of the importance of terminus geometry. Previous limitations on glacier retreat and front geometry are overcome in this algorithm removing technical hurdles preventing such studies being carried out on realistic domains (Cowton et al., 2019). On its own, the calving algorithm is
limited in applying a set melt field to the terminus. Future work should focus on coupling the glacier model, with the calving algorithm, with ocean/plume models (Cook et al., 2023).

## 7.6 Future coupling

The most advanced method of calculating frontal melt in Elmer is the coupled hydrology model developed by Cook et al. (2020). This model couples ice flow with the Glacier Drainage System (GlaDS) module in Elmer/Ice, uses predicted subglacial
meltwater discharge to drive a 1D plume model and determine patterns of frontal melting, and the CD law to predict consequent calving. This work employed the Todd et al. (2018) calving algorithm, and significant further development would be required to reproduce this effort with the new calving algorithm. Future coupling work in Elmer should focus not just on hydrology but also fjord circulation (Cook et al., 2023). The lack of coupling of fjord models with glacier models means there are often large uncertainties when applying melt fields to glacier models. Melt profiles are often derived from buoyant plume theory (Slater
et al., 2017), but the lack of 3D fjord modelling neglects horizontal flow across the front of the glacier. Coupling should aim to be with a high-resolution fjord model such as MITgcm (Cook et al., 2023). Although computationally expensive, it seems futile to solve the glacier dynamics in detail but neglect the same detail with the fjord model. However, many issues such as congruent time-step sizing would need to be resolved.





## 7.7 Future parallelisation

A parallel calving algorithm is currently in the development stage. Conceptually it follows the serial calving routine but undertakes all computationally expensive routines in parallel. In some ways this vastly simplifies the calving algorithm as calving is always implemented in parallel rather than switching between serial and parallel routines. This cuts out the need to gather and redistribute the mesh along with reducing the complexity of the additional new functionality. This would allow increased scalability of the algorithm allowing it to be used in large scale simulations as core models on Elmer/Ice have been shown to 450 scale well up to a thousand cores (Gagliardini et al., 2013).

## 8 Conclusions

The new calving algorithm has been shown to be capable of simulating unrestricted calving and terminus advance. This marks a major step forward in our ability to model and therefore understand calving dynamics. Some improvements to increase the computational efficiency of the algorithm are ongoing but currently the model is robust. Importantly, the new algorithm and its 455 use as part of Elmer/Ice remains computationally light compared to DEMs such as HiDEM and fills the gap between models based on first principles and the widely used SSA-style models.

An assessment of the non-physical parameters and their potential to alter calving, revealed numerical decisions can have a large impact on calving for systems lacking a strong attractor. For systems with strong attractors, such as those with pinning 460 points, the influence of model parameter choice is limited. As such, modellers should be aware of the sensitivity of the system of interest when choosing non-physical parameters. Importantly, calving predicted by the CD law is very coherent when an attractor is present regardless of modelling decisions.

*Code and data availability.* The data associated with this study is made available as supplementary information. The new algorithm including current and future releases is available on the official Elmer/Ice GitHub repository https://github.com/ElmerCSC/elmerfem. The exact 465 code used in this study is available on Zenodo at https://zenodo.org/records/10182705 (Wheel 2023a). The user guides can be accessed from GitHub at https://github.com/ElmerCSC/elmerfem/tree/calving_meshadapt/elmerice/Solvers/Documentation. The three documents associated with the new calving algorithm are Calving3D_lset.md, CalvingRemeshMMG.md and CalvingGlacierAdvance3D.md.

*Video supplement.* Videos of the control experiment for each domain are provided as Supplementary Video 1 and 2.



## Appendix A: Synthetic glacier setup

The synthetic glacier domain extends 5km upstream from the calving front and has a terminus width of 3km (Fig. A1). It flows through a narrowing fjord that has an upstream boundary width of 5km. The fjord geometry projected beyond the initial domain has parallel sidewalls. An initial 3D tetrahedral mesh was created using the meshing software Gmsh. The mesh consisted of 5 layers of 100m resolution that were extruded between surface and bed maps. The horizontal resolution at the terminus was 100m before increasing to 500m at the inflow boundary (Fig. A1 a-c). Two bed geometries were created to produce two

domains, one with a simple downslope and the other with two additional pinning points near the terminus (Fig. A1 d, e). The first domain had the formulation

$$B_h(x,y) = Interior Depth + Width Depth + Terminus Rise, \tag{A1}$$

while the second had the addition to pinning points to give

$$B_h(x,y) = Interior Depth + Width Depth + Bump1 + Bump2 + Terminus Rise, \tag{A2}$$

where

$$Depth(y) = B_0 + y m_b, \ B_0 = -550, \tag{A3}$$

$$Width Depth(x,y) = \frac{|x - 2500|}{(3000 + \frac{y}{5})} \times 400, \tag{A4}$$

$$Bump1(x,y) = exp(H_b \times \frac{(x - 1800)^2 + y^2}{r_b})^2, \ H_b = 100, \ r_b = 550, \tag{A5}$$

$$Bump2(x,y) = exp(H_b \times \frac{(x - 3000)^2 + y^2}{r_b})^2, \ H_b = 100, \ r_b = 550, \tag{A6}$$

$$Terminus Rise(y) = exp(H_t \times \frac{-y}{l_t})^2, \ H_t = 100, \ l_t = 1000, \tag{A7}$$

and where $B_h$ is the bedrock height in meters, $m_b$ is 1/40 the gradient of the bed, $x$ is the x coordinate, $y$ is the y coordinate and $B_0$ is the bed height at $y = 0$. For the bumps, $H_b$ is the height of the bump and $r_b$ is the radius of bump. The surface geometry was identical for both domains and is given by

$$S_h(y) = S_0 + y m_s, \ S_0 = 50, \tag{A8}$$

where $S_h$ is the surface height and $m_s$ is 1/40 the gradient of the surface.

### A1  Boundary conditions

The synthetic glacier domain had six boundary conditions that consist of the calving front of the glacier ($\Gamma_{front}$), both lateral margins ($\Gamma_{left}$, $\Gamma_{right}$), the interior inflow ($\Gamma_{inflow}$), the base ($\Gamma_{base}$) and the top surface ($\Gamma_{surf}$). The boundary conditions applied to each is discussed below.






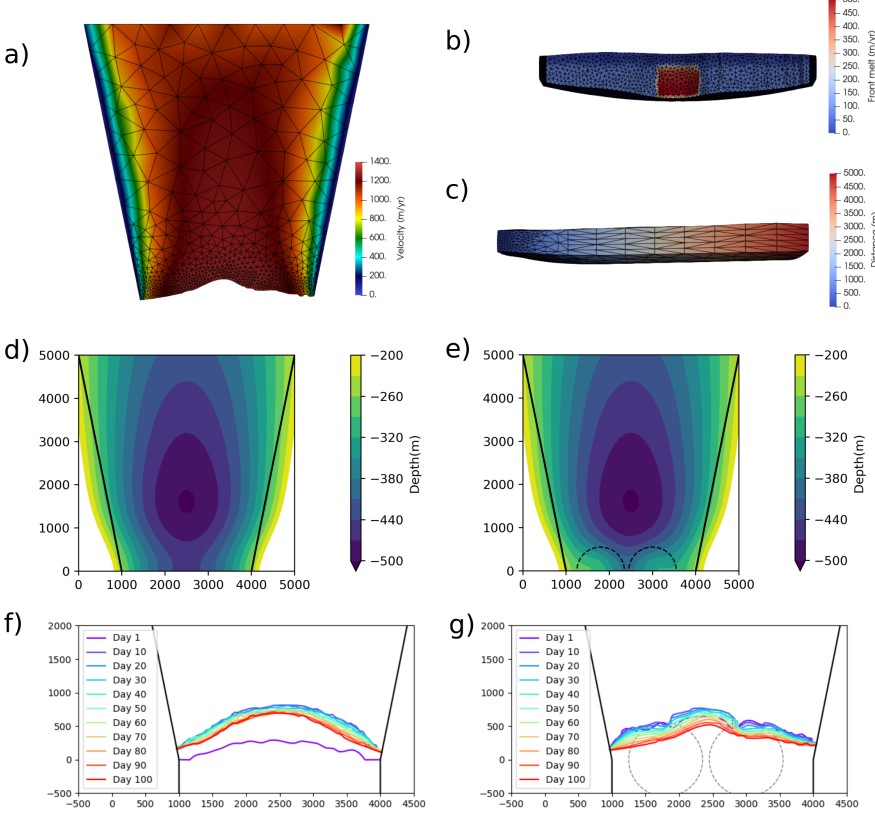

**Figure A1.** The synthetic geometry setup. a) Topdown view of the mesh at the end of the 100d simulation with pinning points. Mesh element size is 50m at the terminus increasing to 500m further upstream. The velocity field is displayed. b) Front view of the from the same simulation showing the horizontal and vertical element size of 50m at the terminus. The melt plume can be seen in the centre of the terminus. c) Side view of the same simulation showing the 50m mesh density at the terminus and 500m at the inflow boundary. d) The bed elevation with no pinning points. e) The bed elevation with pinning points near the terminus. The terminus evolution of during 100d simulation with f) no pinning points and g) pinning points. The full control simulations for each domain can be seen in Supplementary Videos 3 and 4.

Since the lateral boundary represents the fjord walls, a no-penetration condition is implemented at the sidewall margins of the model domain. A simple linear friction law, which has a constant slip coefficient ($\beta$) of 1e-2 $MPa\,m^{-1}\,a$ was applied as a Neumann boundary condition. Therefore, the lateral boundary conditions are

$$u_\perp = 0, \qquad \text{on } \Gamma_{left}, \Gamma_{right}, \tag{A9}$$


$$\sigma_{||} = -u_{||}\beta, \qquad \text{on } \Gamma_{left}, \Gamma_{right}, \tag{A10}$$



where $u$ is the velocity component, $\sigma$ the stress component and the perpendicular and tangential components are shown by $\perp$ and $||$ respectively. A non-linear Weertman friction law is applied to the base with a slip coefficient of 1e-4 $MPa\,m^{-1/3}\,yr^{1/3}$ and an exponent of 3. The base boundary condition is complicated by the grounding line dynamics of the glacier. The grounding of the glacier is solved as a contact problem following Favier et al. (2012). If the glacier is grounded the boundary condition is similar to the lateral boundary with a non-penetration condition applied:

$$u_\perp = 0, \qquad\qquad\qquad \text{on } \Gamma_{base}, \tag{A11}$$

$$\sigma_{||} = Cu_b^{1/m}, \qquad\qquad\qquad \text{on } \Gamma_{base}, \tag{A12}$$

where $C$ is the Weertman slip coefficient, $u_b$ is the basal velocity and $m$ is Weertman exponent. If the glacier is ungrounded, no friction is applied and the glacier is free to move vertically:

$$\sigma_\perp = \min(-\rho_w gh, 0), \qquad\qquad\qquad \text{on } \Gamma_{base}, \tag{A13}$$

$$\sigma_{||} = 0, \qquad\qquad\qquad \text{on } \Gamma_{base}, \tag{A14}$$

where $\rho_w$ is the density of the water, $g$ is the gravitational acceleration and $h$ is the depth below the water level. Similar to ungrounded ice, no friction is applied to the glacier calving face. Since it is in contact with the fjord water body a normal stress is applied below the water level.

$$\sigma_\perp = \min(-\rho_w gh, 0), \qquad\qquad\qquad \text{on } \Gamma_{term}, \tag{A15}$$

$$\sigma_{||} = 0, \qquad\qquad\qquad \text{on } \Gamma_{term}. \tag{A16}$$

The inflow boundary has a fixed velocity of 1000m/yr and this is assumed to be constant throughout the vertical ice column. Therefore, the inflow boundary condition is simply

$$|\boldsymbol{u}| = u_{in}, \qquad \text{on } \Gamma_{inflow}, \tag{A17}$$

where $\boldsymbol{u}$ is the velocity vector and $u_{in}$ is 1000 m/yr. The surface boundary condition is stress-free and surface mass balance is not considered. A submarine melt condition is added to the glacier front boundary or calving front ($\Gamma_{term}$) where a central plume is present along with background melt (Fig. A1 b). The plume is taken from the summer plumes modelled by Kajanto et al. (2023) at Illulissat Fjord. The plume profile was then normalised given the much lower maximum flow velocity at the synthetic geometry of 1500m/yr compared to the much larger speed seen at Jakobshavn Isbrae ($> 10,000$m/yr). Maximal plume melt was set to 500m/yr and the profile adjusted accordingly. No melt was applied to the floating ice on the base.

## A2 Ice properties

A constant temperature of $-20°C$ is set throughout the domain and the associated ice properties are based on Glen's flow, which is then used to solve the full-Stokes equations (Cuffey and Paterson, 2010). The feedback of the temperature dependency of Glen's flow law is calculated using the Arrhenius equation and the rate factors are detailed in (Cuffey and Paterson, 2010).



## A3    Model solvers and parameters

For the transient simulation, the model is run forward for 100 days at timesteps of 1d, giving a total of 100 timesteps. The full-Stokes flow is solved and from this solution the Cauchy stress tensor across the domain is computed. Although there are no surface mass balance conditions, the surface and base free surfaces are solved as a kinematic boundary condition so that the glacier can evolve in response to the flow solution. Similarly, the new front advance routines outlined are used to predict the advance of the glacier down the fjord. As a consequence, the mesh is deformed twice, first vertically and then longitudinally. The longitudinal mesh deformation is limited to 1500m from the calving front of the glacier.

The new calving algorithm as outlined in the main section is applied and the front 1500m of the glacier is remeshed at each timestep. The anisotropic remeshing metric had a minimum horizontal resolution of 50m, which increases to 500m further inland. It has a constant vertical resolution of 50m. The adaptive time stepping present in the calving algorithm was active, with a maximum number of added timesteps set to three. Timesteps were only added if large calving events occurred to capture any consecutive calving from changes in the domain). The calved iceberg threshold was set to 1e7 cubic meters for the adaptive time stepping to be activated. The 2D PlaneMesh that crevasses were mapped onto had a grid size of 40m.

The crevasse depth (CD) law is modified to make the setup more sensitive to calving and highlight potential numerical influences on calving.This is achieved by reducing the *crevasse penetration requirement* to 92.5%. Here, full thickness calving occurs when 92.5% when either: 1) a surface crevasse penetrates 92.5% of the ice column between the surface and water line or 2) surface and basal crevasses extend 92.5% of the entire water column. Although it is known that CD law can underestimate calving (e.g., Choi et al., 2018; Todd et al., 2019; Cook et al., 2023) this should not be concluded in this instance. Instead, the alteration of the calving law should be considered as an increase to the sensitivity of a synthetic glacier so that the effects of varying parameters on calving dynamics can be clearly identified in the following simulation tests. No conclusion on the accuracy of the CD law can be made for a synthetic scenario.

*Author contributions.*  IW wrote the algorithm with technical support from JT and TZ. IW designed and analysed the experiments with guidance from DB and AC. IW wrote the manuscript with contributions from DB and AC. All authors contributed and approved the final manuscript.

*Competing interests.*  The authors declare that they have no conflict of interest

*Acknowledgements.*  This work is from the DOMINOS project, a component of the International Thwaites Glacier Collaboration (ITGC). Support from the Natural Environmental Research Council (NERC: Grant NE/006605/1 ). ITGC Contribution No. ITGC-116. Research



was supported by the HPC-Europa3 program, part of the European Union's Horizon 2020 research and innovation programme under grant

agreement No.730897. TZ was supported by the Finnish Academy COLD consortium grant 322978.



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
