# Peer review of "A new 3D full-Stokes calving algorithm within Elmer/Ice (v9.0)"

_EGUsphere, 2023_

## Author Response (AR1)

Response to Reviewer 1

We thank the reviewer for their helpful comments that will help improve the manuscript. Based on their concerns and those of the second reviewer we have undertaken a reorganisation of the manuscript. The reviewer's comments are highlighted in blue with our response and how corresponding changes to the manuscript below in black.

Major concerns.

My main concern is the need to improve the quality of this manuscript. The authors have used non-scientific language, such as 'almost certain' and 'roughly', when drawing conclusions. Additionally, I have noticed that many introductory contents are misplaced in later sections, such as 4.3, 7.1, 7.5, etc. I suggest that the authors rewrite the introduction as well as these sections. Overall, there are several redundant paragraphs in this manuscript that could be significantly shortened.

We thank the reviewers for their feedback and along with a similar comment from Reviewer 2 we have reworked the manuscript to improve the scientific language and the general structure. Specifically, areas of 'introductory material' have been moved to the introduction or removed. We have also tightened up the manuscript by removing non-scientific language.

Regarding the numerical experiments, particularly the mesh resolutions experiments, it is conventionally recommended to test at least 4 different points of each variable in order to draw conclusions related to convergence. These points should span at least 1/2 and 2 times the control resolution.

We have conducted extra experiments in order to fulfil the requirement to have four different points for each variable spanning half and two times the control resolution. The results of these can be seen in Table 1. Additionally, the terminus positions and calving statistics of these new simulations have been included in the supplement.

The significant impact of non-physical parameters on the no pinning point cases appears to be closely related to the CD calving law. I suggest that the authors add a comparison experiment with a rate-based calving law to identify the reason for this strong dependence.

Yes, the reviewer is correct the crevasse depth (CD) law determines the state of the glacier in the two setups. We have added supplementary experiments with a simple rate-based law to distinguish the difference between the method and the CD law (Appendix B). Predicted retreat and calving varied little between the rate-based experiments with a mean of 4.06m/d. Maximum and minimum values were only 0.01m/d greater or less than the mean (Table B1). The terminus positions and calving statistics of the rate-based experiments have been added to the supplement. We have also changed the terminology used in the paper from 'non-physical parameters' to 'numerical model parameters'.

I fully agree that the authors should focus on the key methodological choices and model capabilities. However, as a whole paper, the manuscript should be self-explanatory without requiring readers to read the supplementary materials.

The reviewer notes correctly that the manuscript has a large supplementary text. Complete documentation of the algorithm requires a large amount of detail that would be accessible to only a small number of the glaciological community interested in complexities of remeshing. On the other hand, as noted by Reviewer 2, it is important that full documentation is available to allow other

glacier modellers to implement the new calving method. We feel that the best compromise was to confine the description of the detailed model algorithm and model setup to the supplement, and to focus the main text on outlining the main features and capabilities of the algorithm, thus allowing the paper to be comprehensible without being burdensomely long.

Detailed comments.

We thank the reviewer for their careful reading of the text. All minor and specific changes requested have been made to the manuscript. Comments that required a fuller explanation are below.

Figure 1. The texts in the color bar are too small to read

Figure 6, the fonts in the figures are too tiny

We have increased the text size in the all the figures presented in the paper including those in the appendix..

Section 4.2, in general, this section need a bit more work.

We have reworked this section as the reviewer suggests.

l132-141, I'm a bit confused about this paragraph. What boundary condition is apply at the fjord wall? If the velocity is solved with a Dirichlet boundary condition parallel to the fjord wall, then the velocity component in the direction of the fjord wall is automatically constrained. Shouldn't the velocity solution be constrained first, instead of fixing the displacement?

The reviewer is correct the velocity is constrained first and then the displacement fixed along a predefined set of fjord walls. We have updated this paragraph for to improve the clarity.

l148-149, 'the model has the …. lateral margins', could you explain a bit more in details, what are the 'boundary elements' and where is the lateral margins? Are these at the calving front, or at the fjord wall?

The lateral margins are those present on the fjord wall so those on the side of the glacier. Boundary elements within the 3D domain are triangular elements present on a particular boundary. The identity of the boundary (and corresponding conditions) enforced on the element are determined using a boundary tag associated with the element. Each boundary element corresponds to a parent tetrahedral element present on the bulk (3D element forming part of the domain). Here, when the glacier advances the Lagrangian movement of the terminus of the glacier mean the calving front can come into contact with the predefined fjord walls. If this occurs, the model transfers the assigned boundary identity of the element from the terminus to the lateral (right or left) boundary. This changes the boundary conditions applied at this element from those present on the terminus (Eqs. A9 and A10) to the lateral boundary (Eqs. A15 and A16) which has a non-penetration condition. We have rewritten this section to provide additional clarity.

l153, is the normal to the element faces, or to the nodes?

Normal to the nodes based on the mean normal of adjacent elements.

l159-161, I'm a bit worried about the melting applied at the corner element, as the normal direction is not going to parallel to the wall, which add additional 'leak of mass' at the side wall boundary

Melt is accounted for prior to terminus advance being reprojected along the side wall boundary. The resultant vector of the velocity vector minus the melt normal vector is constrained along the

predefined fjord wall. The change in mass from melt is usually small compared to the velocity being projected along the fjord walls. Changes in mass from either are insignificant on a glacier scale and when compared to changes in the domain from the Hausdorff distance associated with remeshing.

Overall, it would beneficial to have a schematic plot of the boundary at the corner between ice front and fjord wall, and refer to the items in the figure when explaining the implementations

There is a figure (Fig. 1) in the detailed supplement that provides an example of when boundary elements are transferred from the terminus to the lateral boundary. This has not been included in the main text for brevity as we feel it is only interests those keen in using the detailed modelling methods who will need to read the supplement rather than the general glaciology modelling community. This could be moved into the main text if the reviewer feels this of particular importance.

l210-211, what is 'level set implementation fails across the range of input parameters provided'? Give a concrete example. Why 'calving cannot occur' then?

Given the complexity of remeshing a new internal boundary failure can occur. In order to increase the robustness of the model multiple remeshing parameters can be specified on the model input file allowing several attempts at remeshing and increasing the change of success. This is shown in Figure 4. We have reworked this paragraph to improve clarity.

l219, how is the new solution checked, to ensure not to use unrealistic velocity?

An additional solver that checks the convergence of the velocity solution, the maximum velocity and the divergence from the previous timestep is used. We have added a sentence to describe this process.

l227, why 'plus one second'? Does this mean one has to always use time step at one second?

Yes, as the reviewers note this is currently hardcoded but could easily be changed so this additional time can be user specified. This functionality was included because when an unrealistic velocity is produced the timestep needs to be rerun. However, because of some internal workings in Elmer the time-dependent solvers cannot be rerun with the exact same starting time. One second is insignificant time span when considering the usual time stepping using glacier models. If the reviewer thinks it is particularly important to allow this time to be user defined we can update the code.

Figure 4, row 5, the center box 'Success? Mesh quality sufficient?' has two 'Yes' arrows, which way should 'Yes' go?

Yes, goes two ways. The output calving statistics are only calculated upon successful remeshing. The other 'yes' is the continuation of the algorithm to redistribute the mesh.

l269-274, I could not fully agree with the author's arguments here. Every numerical method for time dependent problem has numerical errors associated with the time scheme. No matter what type of calving law, the errors are due to the approximation of the time derivative in the time dependent equations. In this case, it comes from the free surface equations. I agree the way no-physical rate-based law update the ice geometry is different from rate-based law, but this does not lead to the conclusion that it is time step independent.

We thank the reviewer for this insight. We agree with the reviewer's conclusion that ultimately even rate based laws can be timestep dependent. However, we were trying to emphasise the clear

distinction that the CD law predicts attractor positions towards which the terminus will migrate, and consequently is sensitive to the choice of timestep when the glacier is in a transient state (moving between attractors). This consideration is not an issue for rate-based laws. We have reworked this paragraph to reflect the reviewers comments.

section 6.2, after reading this section several times, it is still not clear to me what adaptive time stepping method is used in this study. To me understanding, adding small timesteps is just a safeguard after the adaptive time stepping. I suggest the authors to rewrite this section, and spend the first few paragraphs to explain what adaptive time stepping method they used in this work.

l303-306, in most numerical models, the common reason to use adaptive time stepping is to improve efficiency of the transient simulation, while maintaining desired accuracy. In general, adaptive time stepping method is more efficient than constant time stepping in terms of getting the final solution of a long term simulation. I would strongly recommend to revise this section.

We thank the reviewer for the above two comments which provide us with us the chance to improve the clarity of the description of the adaptive time-stepping implemented in the model. Firstly, as the reviewer correctly notes the time is reverted if the velocity solution is inadequate. This is not what we describe as 'adaptive time stepping'. The adaptive time stepping is instead a method to better simulate rapid ice loss from a glacier. If a large calving event occurs the new geometry can potentially be unstable. In order to capture secondary calving the time step size is altered if calving occurs over a given threshold. If the calving volume is below the given threshold normal time stepping is resumed. Importantly, this is independent of remeshing and numerical requirements. Remeshing failure (including that of inadequate velocity solution) suppresses calving so normal time stepping must be resumed. Therefore, if remeshing failure occurs it prevents potential calving cascades from being simulated. We have rewritten this paragraph to improve the clarity of this method.

Response to Reviewer 2

We thank the reviewer for their comments that will help improve the quality of the manuscript. The reviewer's comments are highlighted in blue with our response and how corresponding changes to the manuscript below in black.

**Major comment 1: numerical experiments**

The model capabilities are demonstrated with two closely related sets of experiments: they differ in that experiment 1 (unpinned) has no pinning point and experiment 2 (pinned) has two. Here, a pinning point is a local rise in the bedrock where the glacier front will tend to less motion. These are idealized experiments, and well chosen to demonstrate the ultimate capabilities of the model. However, the results are not sufficient to demonstrate convergence with mesh spacing, and clearly show that the results do *not* converge with decreasing time step in either the pinned or unpinned case. The authors do note the lack of convergence in the unpinned case where it is most obvious.

Looking at table 1 (a summary of all experiments), we see that as the time step decreases (2d->1d->1/2d-1/4d) the retreat rate in the pinned case (BTW the column labels are incorrect in table 1) follows the sequence 2.51,3.31,2.99,3.67. This is not convergent: the difference between successive elements is not decreasing. This might improve with yet smaller time steps. In the unpinned case (which is at least as likely in real glaciers as the pinned case) the sequence is clearly diverging. As it stands the method cannot be used with any confidence.

As for space convergence: pick four resolutions following a geometric sequence for each case (e.g 80,40,20,10 m). Then correct (or not correct) behaviors will be evident. It does look from the figures presented as though the unpinned case will not be convergent but the pinned case might be.

Many authors would hide these flaws (or not check at all) and I don't think they are fatal for the paper, but further experiments could show why they occur. The text hints (and I think is probably correct) that the crevasse depth law is the cause rather than the remeshing procedure per se. But this can be tested: carry out simulations with a simple calving rate.

Firstly, to clarify the argument we put forward in the paper, when we discuss 'convergence' we are specifically talking about the predicted changes in terminus position that stem from the calving law. This is not the same as numerical convergence. In the manuscript, we made a conscious effort to discuss and demonstrate the utility of the crevasse depth (CD) law as it represents one of the leading options for a universal calving law. The effects of a position based calving law are rarely discussed in the literature and unlike rate-based laws cannot be related to the numerical convergence of the velocity solution. As the reviewer notes, this distinction between the effects of the calving law and potential numerical convergence issues was not clear in the manuscript.

The crevasse depth calving law predicts the location of attractor points within the advance and retreat system of the glacier. Therefore, when a glacier is in transient state (i.e., the case with no pinning points) the rate of change can be altered by unphysical parameters such as timestep. However, when a clear attractor is present the terminus centres around this point. When altering the unphysical parameters, we described this as converging.

Following the reviewer's comments, we have clarified the distinction between the effects of the calving law and the novel algorithm for implementing any law. An additional set of experiments was run using a rate based calving law of the form:

$$C = u_s - \hat{R},$$

where **C** is the calving rate, $\boldsymbol{u_s}$ is the surface velocity vector at the terminus and $\hat{R}$ is the prescribed scalar retreat distance normal to the terminus. Additionally, experiments have been conducted using this rate-based law with an R value of 1500m/yr. These experiments clearly show that there is limited discrepancy introduced through the remeshing methods (Fig. 1 and Table 1). The algorithm thus exhibits numerical convergence, but solutions based on the CD law can exhibit parameter-related variance in some circumstances. We have taken care to make this distinction clear in the revised text. The results of the additional experiments using the rate-based law will be provided in the supplement of the revised manuscript.

Predicted retreat and calving varied little between the rate-based experiments with a mean of 4.06m/d. Maximum and minimum values were only 0.01m/d greater or less than the mean (Table B1). The rate of retreat throughout the experiments was consistent and did not vary between the two different setup domains where the presence of pinning points had no influence on predicted calving (Fig. B1).

[Figure]

[Figure]

**Figure B1**. Terminus positions through time using a rate-based calving law where the retreat has been specified as 1.5km/yr on a) the domain with no pinning points and b) with pinning points. The outline of the pinning points is shown by the dashed circles.

**Table B1**. Summary of the mean retreat rates for 100d experiments testing the numerical model parameters using a rate-based calving law. The numerical model parameters tested match those in the main text.

| Numerical model parameter | Experiment | Mean retreat rate (m/d) | |
|---|---|---|---|
| | | No pinning points | Pinning points |
| Control | | 4.07 | 4.06 |
| Timestep | 0.25d | 4.06 | 4.06 |
| | 0.33d | 4.06 | 4.06 |
| | 0.5d | 4.07 | 4.06 |
| | 2d | 4.07 | 4.06 |
| Mesh resolution | 25m | 4.07 | 4.06 |
| (both) | 30m | 4.06 | 4.06 |
| | 70m | 4.07 | 4.06 |
| | 100m | 4.06 | 4.06 |
| (vertical) | 25m | 4.05 | 4.05 |
| | 30m | 4.06 | 4.05 |
| | 70m | 4.06 | 4.06 |
| | 100m | 4.05 | 4.06 |
| (horizontal) | 25m | 4.07 | 4.06 |
| | 30m | 4.06 | 4.06 |
| | 70m | 4.06 | 4.05 |
| | 100m | 4.06 | 4.05 |
| Adaptive time | 0 timesteps | 4.07 | 4.07 |
| | 1 timesteps | 4.06 | 4.06 |
| | 5 timesteps | 4.06 | 4.06 |
| | 10 timesteps | 4.06 | 4.06 |

Major comments 2: presentation

I found the paper quite disorganized, at multiple levels. In my opinion it requires a wholesale rewrite.

We have reorganised the paper as requested by the reviewer. As also requested by Reviewer 1, we have moved areas of 'introductory material' to the introduction or removed them entirely.

Many parts of the text are difficult to understand, they are usually descriptions of some model behaviors or feature without examples of quantification, so as a reviewer I am not able to say whether they are likely to be correct or not. This is particularly acute in section 4.4, where numerous algorithmic details are mentioned but it is not clear how they are implemented – following this paper to implement the ideas in (say) ISSM would be impossible.

The reviewer notes an issue we have struggled with when choosing the best way to present the paper. We felt the best compromise was to provide the full algorithm details as a supplement (all 53 pages) which is available outlining everything needed to implement the method within another model. This supplement is too long and method heavy to be of interest to the vast majority of the community and so most of this material has been omitted from the main text. We felt it was more important to show an analysis of the CD law, because this is the only calving law previously used in detailed 3D modelling studies but limited sensitivity testing had been performed. As the reviewers

note there may be some confusion created by reference to details only outlined in the supplement and we have tightened up the main text accordingly.

Figure 3 is overall a very useful figure, providing a set of diagrams that explain the whole procedure well.  It does have a minor flaw: there are no scales and figures are presented at different scales. I can make out what is going on in each case, but published figures should be made with more care. The color scale legend labels in panels a and h are too small.

The figure has been adjusted to include scales for each panel and the colour legends have been made larger.

---

## Author Response (AR2)

We thank the reviewer for their comments. We agree with their comments and have changed the manuscript accordingly. Please see below for the detailed response. Additionally, we have changed the supplement text, so the figures read S1, S2 etc. as requested in the file validation process.

**Main Comments**

- One of the main findings of this manuscript is that the numerical model parameters can significantly impact the results in the case without pinning point, which serves as a crucial warning to the modeling community. I suggest the authors rewrite the abstract to better highlight this finding.

We have added a sentence to the abstract to cover the point the reviewer suggests.

**Detail**

- l62-64, list the names of the numerical model parameters considered in the manuscript

We have added a sentence outlining the numerical model parameters.

- l141, affect → effect Changed

- l148-149, how many degrees of freedom do you have in this case? and how many time steps? These will give the readers a rough idea about what '45 minutes' means. Change 'uniformly' to 'linearly'.

We have added in the maximum DOFs (Stokes solver has the highest DOFs) which is 35,980 (4*number of nodes) for the final domain. Note the DOFs change after remeshing so we have reported the final domain. The number of timesteps for the control of 106 (100 day timestep plus six adaptively added) has also been included.

- l149, exponentially sounds really bad in terms of scaling in numerical method. Do you mean 'cubically' or quadratically'? or something else

We meant cubically as the reviewer points out.

- l173, 'direction of the fjord wall', is this the normal direction? or tangent? Note that the fjord wall is a 2D surface in 3D.

Tangential direction. This has been added. We have added a sentence specifying that the fjord walls are only represented in the x and y plane rather than a 2D surface.

- l279, 'The convergence of the velocity solution' (ln 256) Changed

- l288, it is unclear what 'inadequate flow solution' is in reason 1)

'Inadequate flow solution' has been changed to 'non-converged or unrealistic flow solution'.

 - l287-290, the two reasons are both to reduce the time step. How to adaptively increase dt is not mentioned.

The timestep is only adaptively reduced to assess the potential for a calving cascade or because of an issue requiring remeshing. If these conditions are not met normal time stepping is resumed. We have added a sentence to provide additional clarity.

 - l327, to choose Changed

 - l353, n^2 to n^4 is quadratic to quartic, not exponentially. And, you don't want to have exponential growth of computational cost in general

We meant cubically as the reviewer points out.

 - Appendix A, the words in the equations (A1)-(A7) should be in normal font, not italic

Changed